

# Development and validation of the dizziness fear-avoidance behaviours and beliefs inventory for patients with vestibular disorders

Roy La Touche[1,2,3], Rodrigo Castillejos-Carrasco-Muñoz[4], María Cruz Tapia-Toca[4], Joaquín Pardo-Montero[1,3], Sergio Lerma-Lara[1,3], Irene de la Rosa-Díaz[1,3], Miguel Ángel Sorrel-Luján[5] and Alba Paris-Alemany[1,2,6]

[1] Motion in Brains, Centro Superior de Estudios Universitarios La Salle, Madrid, Spain
[2] Instituto de Dolor Craneofacial y Neuromusculoesquelético (INDCRAN), Madrid, Spain
[3] Department of Physiotherapy, Centro superior de Estudios Universitarios La Salle, Universidad Autónoma de Madrid, Madrid, Spain
[4] Instituto de Otorrinolaringología y Cirugía de Cabeza y Cuello de Madrid, Madrid, Spain
[5] Psicología Social y Metodología, Universidad Autónoma de Madrid, Madrid, Spain
[6] Departamento de Radiología, Rehabilitación y Fisioterapia. Facultad de Enfermería, Fisioterapia y Podología. Universidad Complutense de Madrid, Madrid, Spain

Corresponding author
Roy La Touche,
roylatouche@yahoo.es

## ABSTRACT

The purpose of this study is to present the development and analysis of the factorial structure and psychometric properties of a new self-administered questionnaire (Dizziness Fear-Avoidance Behaviours and Beliefs Inventory (D-FABBI)) designed to measure fear-avoidance behaviors and cognitions related to dizziness disability. A mixed-method design combining a qualitative study with an observational and cross-sectional study was employed to develop (content validity) and psychometrically validate (construct validity, reliability, and convergent/discriminant validity) a new instrument. A total of 198 patients with vestibular disorders (acute vestibular syndrome (AVS), 23.2%; chronic vestibular syndrome (CVS), 35.4%; and episodic vestibular syndrome (EVS) 41.4%) were recruited. Sociodemographic characteristics, the Dizziness Handicap Inventory (DHI) and the Hospital Anxiety and Depression Scale (HADS) and D-FABBI were evaluated. The final version of the D-FABBI consists of 17 items distributed across two subscales: activities of daily living fear-avoidance and movement fear-avoidance.

The D-FABBI showed high internal consistency (Cronbach $\alpha$ = 0.932; 95% CI [0.91–0.94]) and so did the subscales (Cronbach $\alpha$ > 0.8). The exploratory structural equation model and confirmatory factor analysis provided better fit results, with a comparative fit index and root mean square error of approximation values of 0.907 to 0.081. No floor or ceiling effects were identified. There was a positive, significant, and moderate-strong magnitude correlation with the total DHI (r = 0.62) and low-moderate with respect to the HADS depression (r = 0.35) and HADS anxiety subscales (r = 0.26). The patients with CVS had a higher D-FABBI score than those with AVS or EVS.

The D-FABBI appears to be a valid and reliable instrument for measuring the fear-avoidance behaviors and cognition related to dizziness disability of patients with vestibular disorders.

## INTRODUCTION

Dizziness has been defined by international expert consensus as "the sensation of disturbed or impaired spatial orientation without a false or distorted sense of motion" (*Bisdorff et al., 2009*). Dizziness is a common symptom in the adult population, with the estimated lifetime prevalence of significant dizziness ranging from 17% to 30% (*Murdin & Schilder, 2015*), and it has been described as the most frequent reason for otorhinolaryngology consultation (*Guerra-Jiménez et al., 2017*). Studies have also shown an association with functional disability in the elderly (*Aggarwal et al., 2000*; *Dros et al., 2011*; *Mueller et al., 2014*). Moreover, dizziness-related disability leads to greater functional impairment in adults (*Whitney et al., 2004*).

The perception of dizziness involves central and peripheral mechanisms. Processes of habituation, relearning, adaptation, orientation control, and coping might be involved in the onset and maintenance of dizziness (*Yardley & Redfern, 2001*). Psychological factors, specifically cognitive, behavioral and emotional responses, have been suggested as having an important effect on the perception of dizziness (*Yardley & Redfern, 2001*).

Recent studies have shown that dizziness-related disability is not correlated with objective deficits in vestibular clinical tests but have found an association with psychological factors such as anxiety, depression, illness perception, and cognitive and behavioral responses (*Herdman et al., 2020b*, *2020a*; *Wolf et al., 2020*). Other variables that have been shown to be associated with dizziness include fear of motion (*Cuenca-Martínez et al., 2018*; *Grande-Alonso et al., 2018*; *Pinheiro et al., 2021*) and pain catastrophizing (*Cuenca-Martínez et al., 2018*). Although fear and anxiety have been postulated in a theoretical model as factors that influence the perception of dizziness (*Staab, Balaban & Furman, 2013*), several studies have shown moderate-high associations between perceived dizziness and fear of falling (*Song & Lee, 2020*), findings that at least partially verify this theoretical model. Other studies have observed a positive association between fear of physical symptoms and disability due to dizziness in patients with vestibular neuritis (*Cousins et al., 2017*; *Godemann et al., 2005*).

Activity avoidance is a common behavioral response to the perception of dizziness that can hinder psychological and physiological adaptation processes, and it can lead to a vicious feedback cycle of inactivity due to fear of movement and environments that might evoke dizziness (*Yardley & Redfern, 2001*). In addition to the behavioral response (avoidance) and emotional factor (fear), there is an interaction with possible cognitions (beliefs) that might favor the perpetuation of the disability and the perception of dizziness. Studies have shown that negative beliefs regarding the dangers of dizziness, such as fear of

falling, fainting and loss of control, act as predictors of disability at 6 months (*Yardley, Beech & Weinman, 2001*). Other beliefs have been identified such as fear of serious illness, anticipation of a severe episode, and fear of loss of control, the latter being associated with long-term disability (*Yardley, 1994*). *Wolf et al. (2020)* recently showed that negative beliefs about dizziness are even more significant predictors of disability than anxiety, mood and symptom severity. However, the results should be interpreted with caution, given that this study was cross-sectional (*Wolf et al., 2020*).

Current evidence amply supports the relationship between psychological factors and perceived dizziness-related disability, hence the need for evaluating these variables at the clinical and research levels. Psychological factors such as anxiety, depression, illness beliefs, and cognitive and behavioral responses have recently been shown to be associated with disability and symptom severity, with an explained variance of between 30% and 53%, respectively (*Herdman et al., 2020b*), with fear avoidance the only adjusted factor that correlated with the severity of dizziness symptoms (*Herdman et al., 2020b*).

There are very few self-report instruments that measure fear-avoidance beliefs exclusively and specifically for people with dizziness. Indeed, fear-related beliefs have only been assessed with general instruments or instruments designed for other disorders, and they have not been designed and validated psychometrically for patients with dizziness. One of the subscales of the Dizziness Handicap Inventory (DHI) assesses emotional factors involved in disability (*Jacobson & Newman, 1990*), and the Vestibular Rehabilitation Benefit Questionnaire introduced a single item for assessing fear-related beliefs (*Morris, Lutman & Yardley, 2009*).

A recently published study validated the Vestibular Activities Avoidance Instrument (VAAI-9) (*Dunlap et al., 2021a*), a specific questionnaire for measuring fear-related beliefs in patients with dizziness. However, most of the items of this instrument are oriented more towards assessing disability. Thus, the questionnaire showed a very high correlation (0.81) (*Dunlap et al., 2021a*) when convergent validity was assessed with an instrument that measures the disability generated by vestibular disorders in terms of activity and participation (*Alghwiri et al., 2012*), suggesting that the instrument might also be directly measuring the disability construct. Although the VAAI-9 includes four specific items on fear-related beliefs (*Dunlap et al., 2021a*), there are no items that explicitly assess avoidance behavioral responses. There is therefore a need for psychometrically validated self-report instruments that assess avoidance behavioral responses and cognitions associated with fear and avoidance in terms of activity limitations and that also assess the impairments present in patients with dizziness.

The purpose of this study is to present the development and analysis of the factorial structure and psychometric properties of a new self-administered questionnaire (Dizziness Fear-Avoidance Behaviours and Beliefs Inventory (D-FABBI)) designed to measure fear-avoidance behaviors and cognitions related to dizziness disability.

## MATERIALS AND METHODS

The study employed a mixed-method design, combining a qualitative study with an observational and cross-sectional study to develop and psychometrically validate the new

instrument. The design of the Dizziness Avoidance Fear Behaviour and Belief Inventory was developed using a standardized methodology based on six phases (*Artino et al., 2014*): (1) intensive literature review; (2) semi-structured interviews; (3) synthesis of the literature review and semi-structured interview analysis; (4) development of items (writing the items and identifying the domains); (5) expert validation (content validity); and (6) assessment of the instrument's comprehension and feasibility (cognitive debriefing) by a small patient group (pilot testing). The procedures used during the psychometric validation were derived from the COSMIN Study Design checklist for patient-reported outcome measurement instruments (*Mokkink et al., 2010*).

The study was approved by the bioethics committee of the *Centro Superior de Estudios Universitario La Salle* (CSEULS-PI-005/2020). All participants were provided a detailed explanation of the study objectives and gave their written informed consent to participate in the study.

## Development of the items

### Literature review

A review of the relevant scientific literature was performed in specialized databases, and obtained in Medline 29 references, PEDro one reference, PsycINFO 66 references, CINAHL 23 references and EMBASE 130 references. Information related to dizziness-related distress, dizziness-related psychological factors and dizziness-avoidance beliefs and behaviors was extracted. A second search was conducted to identify self-report instruments that aimed to assess dizziness avoidance beliefs and behaviors and psychological factors associated with dizziness disability.

### Semi-structured interviews

Based on the reviewed literature, a semi-structured interview was established for patients with vestibular disorders. The interview was developed by focusing on fear-avoidance beliefs and behaviors and their possible impact on perceived disability. The draft questions were discussed and revised during a pre-arranged supervision session. The semi-structured interview was conducted with 16 patients with chronic vestibular disorders.

### Synthesis of the literature review and semi-structured interview analysis

The results of the semi-structured interview were analyzed using an interpretative phenomenological analysis, as described by *Smith (1996)*. This flexible and versatile qualitative analytical method assesses the meaning that individuals attach to individual experiences. This type of analysis permits participant-centered data to emerge and helps to increase the transparency of the analysis (*Duque & Aristizábal Díaz-Granados, 2019*; *Pringle et al., 2011*; *Tuffour, 2017*).

From the qualitative analysis, an interpretative account of the experiences and a construct of meaning was extracted, and a list of main themes and sub-themes was generated.

The content analysis of the scientific literature was evaluated independently by two researchers (R. L., J. P.) who performed a tabular extraction of the relevant themes. The tables were then pooled and a consensus was reached.

### Development of items

The findings from the relevant literature and the data obtained from the semi-structured interviews were sorted and qualitatively analyzed by four researchers to define the construct concept "fear-avoidance behaviors and beliefs related to dizziness", after which 28 items were designed and subjected to a structured consensus process (*Jones & Hunter, 1995*). Twenty-three items were ultimately established for this phase and ordered by their relevance within each dimension (sub-construct).

### Expert content validity

A preliminary 23-item list was drafted for the scale, whose suitability (relevance, pertinence, clarity, coherence, and degree of coverage of the relevant aspects) was evaluated by an external expert panel (validation by judges).

The content validation panel of experts consisted of 10 expert judges with research and clinical backgrounds (two medical doctors, five physiotherapists and three psychologists) who conducted a qualitative assessment (relevance, comprehensiveness and comprehensibility) of each item using a three-level Likert scale (agree; neither agree nor disagree; and disagree).

To consider an item for deletion, the following performance indicators were evaluated: (1) mean item score of <0.70 for Aiken's V statistic (*Aiken, 1985, 2016*); (2) the behavioral content does not have a generally accepted meaning or definition; (3) the item is ambiguously defined; (4) the content item is irrelevant or repetitive to the purposes of measurement; and (5) whether the judges agreed that the item had been adequately sampled based on consensus.

### Cognitive debriefing

A cognitive debriefing methodology was applied as a qualitative evaluation of the preliminary version of the instrument by a small patient group (29 patients). The cognitive debriefing was based on the evaluation of the instrument considering five aspects analyzing the completeness, relevance and clarity of expression (*Farnik & Pierzchała, 2012*):

1) Comprehension of each question (including the question's intent and meaning). The questions were a) "Did you have any difficulty in understanding the question" and b) "What does the question mean to you?"

2) Relevance of the information. The questions were a) "What does the question mean to you?" and b) "Do you think it is an important question?"

3) Decision processes (response time, response/abandonment rate).

4) The response processes. The question was a) "Do you feel that the answer choices are appropriate and match the content of the questions?"

5) General comments. The questions were a) "What essential aspects of knowledge of dizziness were missing from the instrument?", b) "Do you consider the length of the questionnaire to be adequate?", c) "Do you consider the length of the questionnaire to be adequate?"

## Psychometric validation

### Participants

A consecutive non-probability sample of participants was recruited from a physiotherapy clinic and from an otorhinolaryngology medical clinic specializing in balance disorders. All participants were recruited, diagnosed and classified by an otorhinolaryngologist specializing in vestibular disorders. The vestibular syndromes were categorized according to the international classification of vestibular disorders (ICVD) (*Bisdorff, Staab & Newman-Toker, 2015*): acute vestibular syndrome (AVS), episodic vestibular syndrome (EVS) and chronic vestibular syndrome (CVS).

Patients were selected if they met all of the following criteria: (1) presence of vestibular syndrome, the diagnosis of which was made according to the ICVD (*Bisdorff, Staab & Newman-Toker, 2015*); (2) presence of dizziness as the main vestibular symptom, although it could be accompanied by other symptoms such as vertigo, vestibular-visual symptoms and postural symptoms (*Bisdorff et al., 2009*); (3) at least 18 years of age; and (4) good understanding of Spanish. The exclusion criteria were: (1) cognitive impairment, and (2) presence of psychiatric limitations that would impede participation in the study assessments. The inclusion and exclusion criteria were screed by an Otorhinolaryngology specialist in vestibular disorders, in charge of assessing the medical history of the patients.

### Sample size

The sample size for the psychometric evaluation was established through a theoretical profile based on an exploratory factorial analysis and estimated to exceed 200 cases, based on a moderate condition where communalities of 0.40–0.70 and at least two factors with more than four items each are expected (*Lloret-Segura et al., 2014*). This estimate is in line with the methodological criteria of experts who consider that even under ideal conditions, such as obtaining high communalities and well-determined factors, the sample for studies that conduct a factorial analysis should exceed 200 cases (*Ferrando Piera & Anguiano Carrasco, 2010*; *Lloret-Segura et al., 2014*).

The sample size calculation for the test-retest reliability study used the method described by *Walter, Eliasziw & Donner (1998)*, which is based on estimating the sample size from assumptions of the intraclass correlation coefficient (ICC) result. For the test-retest assessments (two assessments), the minimum acceptable ICC was estimated at P0 = 0.75 (*Koo & Li, 2016*); however, an ICC greater than P1 = 0.90 was expected. Considering a power of 80% (β = 0.2) and an alpha error level of 0.05, the study sample size was calculated at 26 participants. By estimating possible losses of 25% for the sample, the total recommended sample size would be 35 participants. To perform this calculation, we employed the Power Analysis and Sample Size (PASS 12; NCSS Statistical Software, Kaysville, UT, USA) software.

### Procedure

After consenting to participate in the study, the participants received a series of self-reports to assess variables related to disability and emotional status and to record demographic

characteristics. The self-reports included the draft version of the D-FABBI, the DHI and the Hospital Anxiety and Depression Scale (HADS).

The sociodemographic questionnaire collected information on gender, date of birth, marital status, educational level and employment status.

### Dizziness fear-avoidance behaviors and beliefs inventory (Draft version)

The preliminary version of the D-FABBI consisted of 19 items and two theoretical subscales that evaluated (1) the behaviors and cognitions of fear avoidance of movement-related dizziness and (2) the behaviors and cognitions of fear avoidance of dizziness related to functional activities and activities of daily living (ADL). The items are scored on a one- or four-point Likert-type scale (total disagreement, some disagreement, some agreement, or totally agree). Higher scores indicate greater fear-avoidance behaviors and cognitions associated with dizziness.

## Data analysis

For all statistical analyses, we used SPSS software version 21 (IBM SPSS Statistics), and the R packages psych (*Revelle, 2023*), semTools (*Jorgensen et al., 2019*) and lavaan packages (*Rosseel, 2012*).

Descriptive statistics were used to summarize the data for categorical variables as absolute (number) and relative frequencies (percentage). Sociodemographic and clinical variables are presented as mean ± standard deviation (SD), 95% confidence interval (CI) and range (minimum-maximum). A one-way analysis of variance (ANOVA) was used to determine the differences between the D-FABBI and its subscales with respect to the various patient groups. A *post hoc* analysis with a Bonferroni correction was performed in the case of significant ANOVA findings for multiple comparisons between variables. Effect sizes (*d*) were calculated according to Cohen's method, in which the magnitude of the effect was classified as small (0.20–0.49), medium (0.50–0.79) or large (0.8) (*Cohen, 1988*).

### Construct validity

The construct validity was assessed employing a bifurcated procedure: (1) an exploratory factor analysis (EFA) was implemented to ascertain the most fitting factor structure and (2) a confirmatory factor analysis (CFA) was utilized to corroborate the theoretical factor structure of the proposed model.

The factorial structure was investigated using a principal axis factoring with oblimin rotation (*Izquierdo, Olea & Abad, 2014*). The quality of the factor analysis models was assessed using the Kaiser-Meyer-Olkin (KMO) test and the Bartlett sphericity test. The KMO measures the degree of multicollinearity and ranges from 0 to 1 (should be >0.50–0.60) (*Kaiser, 1974*). The optimal number of factors was established based on Kaiser's eigenvalue criterion (eigenvalue ≥ 1) and evaluation of the scree plot (*Ferguson & Cox, 1993*). For the EFA, a factor loading >0.4 was considered necessary for the item's inclusion in each factor (*Guadagnoli & Velicer, 1988*).

We estimated a CFA model with a simple structure and an exploratory structural equation model (ESEM). The CFA structure was constructed according to the theoretical model. This CFA model can be understood as a special case of the ESEM model where

some factor loadings have been set to zero. The ESEM will estimate all the factor loadings, like the EFA, while allowing for the inclusion of correlated errors, if necessary, like the CFA (*Asparouhov & Muthén, 2009*). Accordingly, the ESEM can be more suitable to model complex factor loadings structures and will provide more accurate factor loadings and factor correlations in that situation (*Nájera, Abad & Sorrel, 2023*). Given that multivariate normality was not assumed, we used the robust maximum likelihood estimator. Model fit was assessed using the root mean square error of approximation (RMSEA), and the comparative fit index (CFI). A CFI > 0.90 and RMSEA < 0.08 reflected acceptable fit (*Hu, Bentler & Peter, 2009*). A noticeable decrease in fit when transitioning from the more general ESEM to the CFA can be expected when certain cross-loadings are of non-trivial magnitude in the ESEM (*Garrido et al., 2020*). Factor loadings and factor correlation estimates were then inspected and compared across the multiple models. The CFA model was expected to overestimate the factor correlation in the presence of cross-loadings (*Hsu et al., 2014*). When this overestimation is observed, it is advisable to retain the ESEM model. The model modification indices (MI) at the 99% confidence level were inspected to locate possible sources of model misfit.

### Floor and ceiling effect

The procedure was evaluated by calculating the percentage of patients achieving the minimum or maximum possible scores. If at least 15% of the patients achieved the minimum/maximum score, the floor/ceiling effect was considered to be present (*Terwee et al., 2007*).

### Convergent validity

For the convergent validity using Pearson correlations between D-FABBI and the other dizziness-related disability and psychological measures (DHI and HADS), a value <0.30 was considered a low correlation, 0.30–0.60 a moderate correlation, and >0.60 a strong correlation (*Terwee et al., 2007*).

- Disability associated with the subjective sensation of dizziness was measured with the Spanish version of the DHI, an instrument used as a measure to quantify the impact of dizziness on the patient's quality of life as assessed in various dimensions. This self-report measure consists of 25 items and three factors (emotional DHI, functional DHI, physical DHI). The Spanish DHI has been found to have adequate psychometric properties (*Pérez et al., 2000*).
- Mood was assessed using the Spanish-validated HADS (*Herrero et al., n.d.*; *Herrmann, 1997*), a scale consisting of 14 items subdivided into two sub-scales of anxiety and depression. Each item is scored with four points ranging from 0–3. Final scores <8 indicate an absence of depression or anxiety, scores of 8–10 are inferred as borderline values, while scores >10 are indicative of high levels of this variable (*De Las Cuevas Castresana, Garcia-Estrada Perez & Gonzalez de Rivera, 1995*). The Spanish-validated HADS has good psychometric properties (*Herrero et al., n.d*).

*Reliability*

The internal consistency, test-retest reliability, the measurement error (as standard error measurement (SEM)) and the minimum detectable change were examined and calculated as previously described in *La Touche et al. (2020)*.

*Discriminant validity*

We conducted a discriminant validity analysis of the D-FABBI to determine the various levels of limitation of life activity and movement due to dizziness associated with fear-avoidance behaviors and beliefs. As a criterion variable, we used the levels of disability (functional and emotional) of the DHI questionnaire (0–14 points corresponds to no disability; 15–24 points corresponds to moderate disability; and >25 points corresponds to severe disability).

We employed the analysis described in *La Touche et al. (2020)* to determine the level of limitation and the proportion of patients correctly classified.

We calculated the optimal cut-off point between levels of limitation of life activity and movement due to dizziness associated with fear-avoidance behaviors and beliefs using the Youden index (*Böhning, Böhning & Holling, 2008*). Also the sensitivity, specificity, negative and positive predictive value for each scores were assessed.

## RESULTS

The total sample consisted of 201 participants (three sample dropouts/198 participants ultimately analyzed), of which 70.2% were women. Most patients were diagnosed with EVS (41.4%). The remaining patients had CVS (35.4%) or AVS (23.2%). There were statistically significant differences between the results of the D-FABBI (F = 8.91; $P < 0.001$), the ADL fear-avoidance (F = 9.41; $P < 0.001$) and the movement fear-avoidance (F = 6.20; $P = 0.002$) subscales with respect to the various diagnoses. In the multiple comparisons analysis, there were higher rates for the CVS group than for the AVS group ($P < 0.001$; $d > 0.67$). Table 1 presents the patients' sociodemographic characteristics and scores on the various self-reported scales.

### Exploratory factor analysis

The KMO test showed acceptable data for the factor analysis (KMO score of 0.891), there were no multicollinearity problems, and Bartlett's test of sphericity rejected the identity matrix null hypothesis ($\chi^2$ (171) = 1,609.65, $P < 0.001$).

These results justify continuing with the EFA. Subsequently, we implemented the principal axis method for factor extraction incorporating oblimin rotation. Both the scree-plot, Kaiser's criteria, and the parallel scrutiny involving the polychoric correlation matrix advocated for the retention of two factor (see Fig. 1). Consequently, the entirety of the criteria coalesced around a two-factor solution, jointly accounting for 52.32% of the overall variance. The first factor (46% of the total variance) consisted of nine items. The theoretical contents of this factor refer to the fear-avoidance behaviors to ADL (factor 1 was called "ADL fear-avoidance"). The second factor (6.32% of the total variance) consisted of items 1–4, 7, 8, and 15–18 and was named "movement fear-avoidance" because it mainly focused

**Table 1 Sociodemographic data and scores obtained on the self-reported scales.**

| Sociodemographic and clinical data | Mean ± SD | Range (Min-Max) |
|---|---|---|
| Age (years) | 57.48 ± 15.43 | 17–87 |
| BMI (kg/m$^2$) | 24.82 ± 3.78 | 17.04–37.87 |
| Dizziness fear-avoidance behaviours and beliefs inventory | | |
| ADL fear-avoidance | 19.41 ± 8.23 | 8–40 |
| Movement fear-avoidance | 16.71 ± 5.76 | 6–28 |
| Total scale score | 36.13 ± 12.75 | 16–68 |
| Dizziness handicap inventory | | |
| Functional | 11.30 ± 8.03 | 0–34 |
| Emotional | 8.14 ± 7.33 | 0–32 |
| Physical | 11.56 ± 6.15 | 0–28 |
| Total scale score | 30.83 ± 19.01 | 0–90 |
| Hospital anxiety and depression scale | | |
| Anxiety | 6.70 ± 3.84 | 0–18 |
| Depression | 6.15 ± 2.94 | 0–14 |
| Total scale score | 12.85 ± 6.03 | 2–30 |
| **Categorical variables** | *N (%)* | |
| Patient diagnosis | | |
| AVS | 46 (23.2) | |
| EVS | 82 (41.4) | |
| CVS | 70 (35.4) | |
| Gender | | |
| Women | 139 (70.2) | |
| Men | 58 (29.3) | |
| Employment status | | |
| Employed | 101 (51) | |
| Unemployed | 11 (5.6) | |
| Medical leave due to disability | 15 (7.6) | |
| Retired | 68 (34.3) | |
| Level of education | | |
| Uneducated | 3 (1.5) | |
| Primary education | 19 (9.6) | |
| Secondary education | 45 (22.7) | |
| University education | 131 (66.2) | |

on fear-avoidance behaviors of movement. The factor loading of each item is shown in Table 2.

## Confirmatory factor analysis

Table 3 lists the model fit results. The CFA model did not display acceptable fit levels, indicating that a simple structure model is unsuitable for these data. This was manifested in the presence of four modification indexes referring to cross-loadings: x4, x8, and x17 for
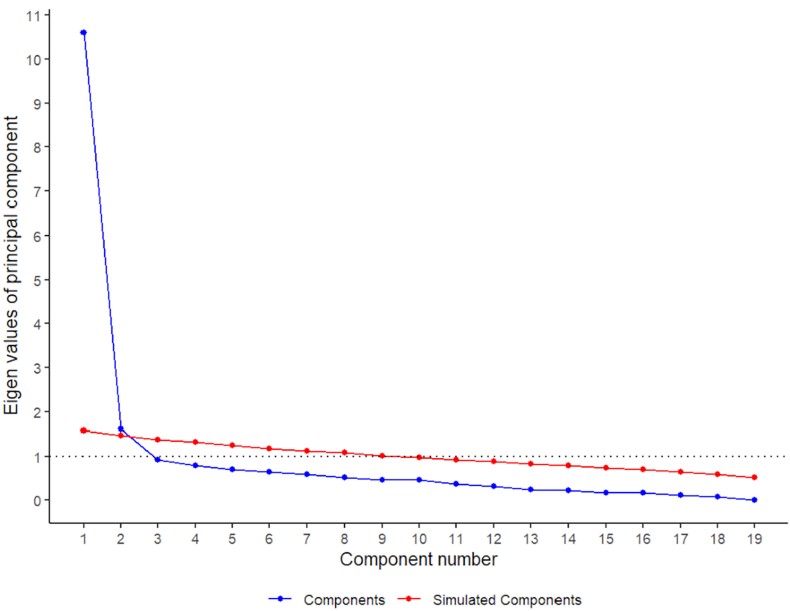

**Figure 1 Parallel analysis results based on the polychoric correlation matrix and the principal components.**

the first factor and x7 for the second factor. There were also eight other significant modification indexes referring to correlations between error terms. Consistent with these results, the ESEM model was found to provide a better fit, with CFI and RMSEA values of 0.081 to 0.907. Two of the items (x4 and x8), which *a priori* were indicative of F2, loaded more F1 and were therefore eliminated. To improve the fit of this reduced model, certain error-term covariances were sequentially released taking the ESEM model as the baseline model. In each iteration, the modification index that reached a larger size was selected, that is, the one that, if included, would reduce the misfit to a greater extent. In total, two additional parameters were necessary to achieve an acceptable fit (*i.e.*, CFI > 0.90 and RMSEA < 0.08). In the first step the error terms for items x2 and x7 (MI = 32.25) and in the second step the error terms for items x6 and x13 (MI = 23.11) were correlated. These model modifications involved items referring to related concepts (*i.e.*, x2: "I'm afraid of moving my neck and head", x7: "I avoid turning over in bed so as to prevent the symptoms from appearing", x6: "I try to walk with short and slow steps" and x13: "Someone with my condition should not be exercising"), which provided certain substantive grounds for their inclusion.

Table 3 lists the model parameter estimates. Although the loadings in the CFA solution were high, the fact that the correlation between the two factors is very high (0.813) and the presence of high cross-loadings in the ESEM solution (*e.g.*, x7, x4, x8) led, as with the results indicated in Table 4, to dismiss the simple structure. We observed that as the modification indices were included, the solution increasingly resembled the theoretical structure. In ESEM MI 2, the remaining items did not have large secondary loadings, except for item x15 that nonetheless had the greatest loading in F2, as was indicated by the theoretical model. We checked that eliminating items four and eight did not lead to a

**Table 2 Items of D-FABBI distribution and factor loadings according to principal axis factoring with Oblimin rotation including Kaiser correction (N = 198).**

| Item | Factor 1 (ADL fear-avoidance) | Factor 2 (Movement fear-avoidance) |
|---|---|---|
| 19. When I go out for a walk, I try to walk in places where I can hold on to something to walk more safely. | 0.81* | 0.59 |
| 5. It's not safe for someone with my condition to walk the street alone. | 0.80* | 0.49 |
| 6. I try to walk with short, slow steps. | 0.79* | 0.58 |
| 10. I avoid physical and sports activity. | 0.77* | 0.54 |
| 13. A person with my condition should not exercise. | 0.74* | 0.46 |
| 9. I'm afraid to drive. | 0.73* | 0.51 |
| 11. Because of my condition, I avoid climbing heights such as stairs and balconies | 0.72* | 0.62 |
| 12. I avoid using computers and tablets. | 0.71* | 0.49 |
| 14. I avoid household activities. | 0.70* | 0.39 |
| 2. I'm afraid to move my neck and head. | 0.43 | 0.82* |
| 17. I avoid turning my head to look to the side. | 0.56 | 0.79* |
| 16. I avoid looking up because it makes me feel sick. | 0.56 | 0.73* |
| 15. I'm afraid to bend down. | 0.64 | 0.68* |
| 7. I avoid turning over in bed so as to prevent the symptoms from appearing. | 0.58 | 0.67* |
| 4. A person with my condition should not make sudden movements. | 0.58 | 0.62* |
| 1. I always think about the movement I'm going to make before I physically do it. | 0.37 | 0.59* |
| 8. I avoid staring at fixed points for too long. | 0.55 | 0.57* |
| 3. I'm afraid of falling down. | 0.35 | 0.55* |
| 18. The dizziness tells me that I shouldn't move. | 0.47 | 0.54* |

**Note:**
* Items included at each of the factors.

relevant reduction in the reliability of the scales estimated with the ESEM model and the final ESEM MI two models (factor reliability with the original 19 items: omega = 0.877 and 0.749; after removing items 8 and 4; omega = 0.859 and 0.744 for factors 1 and 2, respectively). These two factors were positively correlated (0.525).

## Floor and ceiling effect

There was no floor or ceiling effect. Nine patients scored 18 points, which is the minimum possible score (4.55%), and only one patient scored the maximum (0.51%/68 points).

## Convergent validity

In general terms, the ADL fear-avoidance and movement fear-avoidance subscales presented a moderate magnitude relationship with the DHI emotional and physical subscales and HADS-depression subscale. Table 5 shows the correlations between the D-FABBI and its subscales on the one hand and all the assessed self-reported scales on the other.

**Table 3 Model fit.**

| Model | χ2 | df | P-value | CFI | RMSEA |
|---|---|---|---|---|---|
| CFA | 344.858 | 151 | <0.001 | 0.873* | 0.089* |
| ESEM | 282.291 | 134 | <0.001 | 0.907 | 0.081* |
| ESEM (without x4 and x8) | 243.275 | 103 | <0.001 | 0.899* | 0.091* |
| MI1 x2 ~~ x7 | 215.518 | 102 | <0.001 | 0.918 | 0.082* |
| MI2 x6 ~~ x13 | 190.340 | 101 | <0.001 | 0.935 | 0.074 |

Notes:
~~: Correlations between error terms. *CFI < 0.90/RMSEA > 0.08. MI, Modification index; CFA, confirmatory factor analysis; ESEM, exploratory structural equation model; RMSEA, root mean square error of approximation; CFI, comparative fit index.

## Reliability

The internal consistency of the D-FABBI was 0.932 (95% CI [0.91–0.94]), with its two subscales showing an internal consistency >0.800 (movement fear-avoidance: 0.849; 95% CI [0.81–0.88]. ADL fear-avoidance: 0.921; 95% CI [0.90–0.93]). To assess the instrument's test-retest reliability, 35 patients (82.9% women; mean age, 58.23 ± 16.45 years; mean body mass index, 25.31 ± 3.75) re-took the scale a mean of 8.25 ± 1.85 days later. According to the ICC, the scale's stability over time was excellent, with an $MDC_{95}$ of 3.81. Table 6 shows the descriptive statistics and results of the test-retest reliability and responsiveness analysis for the D-FABBI and its subscales.

## Discriminant validity

The ANOVA results show that there were differences in the various fear-avoidance behaviors and cognitions associated with dizziness levels (F = 127.44; $P < 0.001$). In the *post hoc* tests, there were significant differences ($P < 0.001$) between each of the established levels (Fig. 2). The highest percentage of patients was in the subclinical level of fear-avoidance behaviors and cognitions associated with dizziness (38%). Statistically significant differences ($X^2$ = 35.74; $P < 0.001$) were found between the percentages of patients per group and the level of fear-avoidance behaviors and cognitions related to dizziness disability (Fig. 3). Table 7 shows the other percentages and descriptive statistics according to level.

The D-FABBI had good and very good diagnostic accuracy (in terms of specificity and sensitivity, respectively) for classifying patients at the severe level; the classification for the moderate level was sufficient for sensitivity and very good. The optimal cutoff for considering fear-avoidance behaviors and cognitions associated with dizziness was 33 points. Table 7, and Figs. 4 and 5 show the results of the diagnostic accuracy and all optimal cut-off points.

## DISCUSSION

The objective of this study was to develop and psychometrically validate a new instrument that assesses self-reported fear-avoidance cognitions and behaviors in patients with vestibular disorders. The study's findings show that the instrument has good psychometric properties and good sensitivity and specificity in classifying severe fear-avoidance

**Table 4 Model parameter estimates.**

| Item | CFA F1 | CFA F2 | ESEM F1 | ESEM F2 | ESEM (17 items) F1 | ESEM (17 items) F2 | MI2 (17 items) F1 | MI2 (17 items) F2 |
|---|---|---|---|---|---|---|---|---|
| x19 | 0.749 | | 0.637* | 0.146 | 0.627* | 0.165 | 0.568* | 0.221 |
| x6 | 0.784 | | 0.660* | 0.165 | 0.647* | 0.179 | 0.723* | 0.127 |
| x5 | 0.711 | | 0.771* | −0.065 | 0.756* | −0.050 | 0.772* | −0.059 |
| x10 | 0.771 | | 0.835* | −0.066 | 0.828* | −0.053 | 0.802* | −0.024 |
| x9 | 0.746 | | 0.633* | 0.136 | 0.623* | 0.152 | 0.575* | 0.189 |
| x7 | 0.671 | | 0.406* | **0.368** | 0.404* | **0.363** | 0.447* | 0.294 |
| x12 | 0.712 | | 0.637* | 0.106 | 0.627* | 0.121 | 0.608* | 0.148 |
| x13 | 0.697 | | 0.839* | −0.152 | 0.833* | −0.137 | 0.876* | −0.150 |
| x11 | 0.737 | | 0.596* | 0.197 | 0.588* | 0.204 | 0.543* | 0.238 |
| x14 | 0.613 | | 0.722* | −0.124 | 0.721* | −0.107 | 0.726* | −0.100 |
| x1 | | 0.549 | 0.049 | 0.537* | 0.052 | 0.535* | 0.052 | 0.532* |
| x2 | | 0.652 | −0.028 | 0.714* | −0.021 | 0.697* | −0.012 | 0.662* |
| x3 | | 0.533 | 0.058 | 0.484* | 0.063 | 0.470* | 0.074 | 0.457* |
| x4 | | 0.643 | 0.438* | 0.288 | | | | |
| x8 | | 0.603 | 0.415* | 0.274 | | | | |
| x15 | | 0.741 | **0.383** | 0.439* | **0.379** | 0.453* | **0.380** | 0.457* |
| x16 | | 0.727 | 0.065 | 0.700* | 0.061 | 0.710* | 0.049 | 0.726* |
| x17 | | 0.799 | −0.014 | 0.865* | −0.024 | 0.881* | −0.029 | 0.896* |
| x18 | | 0.500 | 0.238 | 0.296* | 0.232 | 0.297* | 0.172 | 0.350* |
| | cor (F1, F2) = 0.813 | | cor (F1, F2) = 0.621 | | cor (F1, F2) = 0.610 | | cor (F1, F2) = 0.603 | |

**Notes:**
The highest values of the ESEM solutions are marked with an asterisk (*) and any cross-loading higher or equal than 0.30 is shown in bold. MI, Modification index; CFA, confirmatory factor analysis; ESEM, exploratory structural equation model.

**Table 5 Convergent validity of the D-FABBI.**

| Convergent validity | D-FABBI Total score | ADL fear-avoidance | Movement fear-avoidance |
|---|---|---|---|
| DHI | 0.62** | 0.61** | 0.49** |
| DHI functional | 0.63** | 0.63** | 0.50** |
| DHI emotional | 0.47** | 0.51** | 0.31** |
| DHI physical | 0.52** | 0.47** | 0.47** |
| HADS | 0.34** | 0,34** | 0.25** |
| HADS-anxiety | 0.26** | 0.25** | 0.21** |
| HADS-depression | 0.35** | 0.38** | 0.24** |

**Notes:**
** $P < 0.001$.
D-FABBI, Dizziness Fear-Avoidance Behaviours and Beliefs Inventory; DHI, Dizziness Handicap Inventory; HADS, Hospital Anxiety and Depression Scale.

**Table 6 Descriptive statistics, intraclass correlation coefficients (ICCs) and associated 95% confidence intervals (CIs), standard error of measurement (SEM), minimal detectable change (MDC90) and MDC95.**

| | Mean ± SD | | ICC (95% CI) | SEM | MDC$_{90}$ | MDC$_{95}$ |
|---|---|---|---|---|---|---|
| | Test 1 | Test 2 | | | | |
| D-FABBI | 46.54 ± 9.83 | 48.28 ± 8.06 | 0.84 [0.71–0.92] | 1.38 | 3.21 | 3.82 |
| ADL fear-avoidance | 26.65 ± 6.31 | 27.31 ± 5.88 | 0.8 [0.63–0.89] | 0.91 | 2.09 | 2.50 |
| Movement fear-avoidance | 19.88 ± 5.03 | 20.62 ± 3.85 | 0.86 [0.75–0.93] | 0.80 | 1.87 | 2.23 |

Note:
D-FABBI, Dizziness Fear-Avoidance Behaviours and Beliefs Inventory; SD, Standard deviation; ICC, intraclass correlation coefficient, model; MDC$_{90}$, minimal detectable change at the 90% confidence level; MDC$_{95}$, minimal detectable change at the 95% confidence level; SEM, standard error of the measurement.

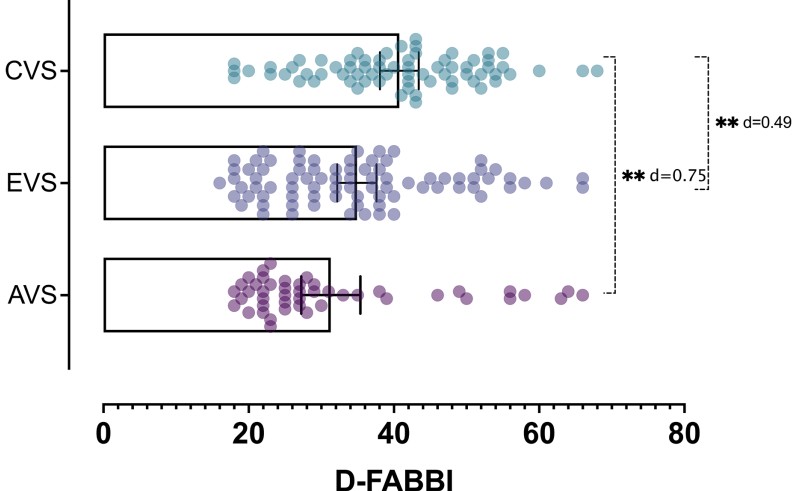

**Figure 2 The differences between the three groups of patients according to the result of the D-FABBI.** Abbreviations: acute vestibular syndrome (AVS), episodic vestibular syndrome (EVS) and chronic vestibular syndrome (CVS). **$P < 0.001$.     

behaviors and cognitions and their relationship with disability. To our knowledge, this is the first instrument of its type to be developed in Spanish and one of the few such instruments worldwide.

The D-FABBI has been rigorously generated through a process that included a qualitative study with patients with chronic vestibular disorders, a literature review and a content analysis of the instrument by an expert committee. The exploratory and confirmatory factorial analysis corroborated the proposed factorial structure, which consists of two factors that assess the cognitions and fear-avoidance behaviors related to movement and those related to ADL.

The two instruments related to fear-avoidance beliefs in patients with vestibular disorders (the VAAI-9 (*Dunlap et al., 2021a*) and the dizziness catastrophizing scale (*Pothier et al., 2018*)) presented a unifactorial structure in their validation. Our results suggest that D-FABBI had a clearly bifactorial structure because its items are more explicitly worded, given that the items used words directly related to the fear-avoidance
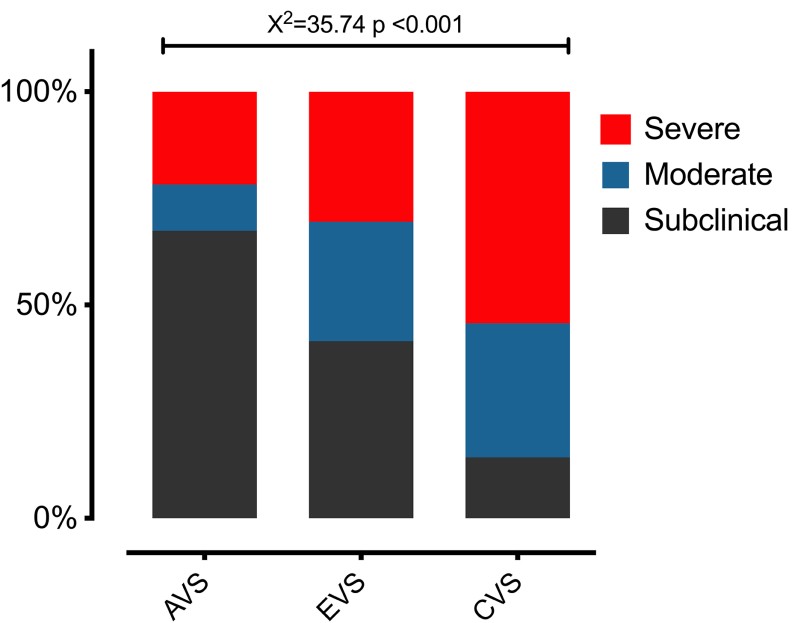

**Figure 3 The differences in the percentages of patients per group and the level of fear-avoidance behaviors and cognitions related to dizziness disability.** Abbreviations: acute vestibular syndrome (AVS), episodic vestibular syndrome (EVS) and chronic vestibular syndrome (CVS).

**Table 7 Diagnostic accuracy results and all optimal cut-off points.**

| Diagnostic accuracy and cut-off points | Subclinical | Moderate | Severe |
| --- | --- | --- | --- |
| Mean ± SD | 25.6 ± 6.27 | 34.96 ± 8.77 | 47.72 ± 10 |
| 95% CI | [24.18–27.07] | [32.46–37.45] | [45.39–50.06] |
| Cases, N (%) | 75 (38%) | 50 (25.3%) | 73 (37%) |
| Optimal cuff-off | <33 | ≥33 | ≥42 |
| Sensitivity (95% CI) | – | 0.68 [0.53–0.80] | 0.76 [0.65–0.85] |
| Specificity (95% CI) | – | 0.86 [0.76–0.93] | 0.82 [0.68–0.91] |
| Positive predictive value (95% CI) | – | 0.77 [0.63–0.86] | 0.86 [0.75–0.92] |
| Negative predictive value (95% CI) | – | 0.80 [0.68–0.89] | 0.70 [0.58–0.84] |

**Note:**
SD, standard deviation; 95% CI, 95% confidence interval.

model such as "avoidance" and "fear", as well as clear examples of fear-avoidance behaviors and cognitions related to dizziness in various movement situations or specific ADL. The dizziness catastrophizing scale is an instrument adapted from the pain catastrophizing scale. Although the construct of catastrophizing has been included among the factors involved in the fear-avoidance model (*Vlaeyen & Linton, 2000*), the items of dizziness catastrophizing scale are not written around fear-avoidance cognitions and behaviors. The VAAI-9 (*Dunlap et al., 2021a*) is a specific instrument on fear-avoidance beliefs and behaviors for vestibular disorders; unlike the D-FABBI, however, half of the items in the VAAI-9 are more related to disability than to fear-avoidance behaviors.

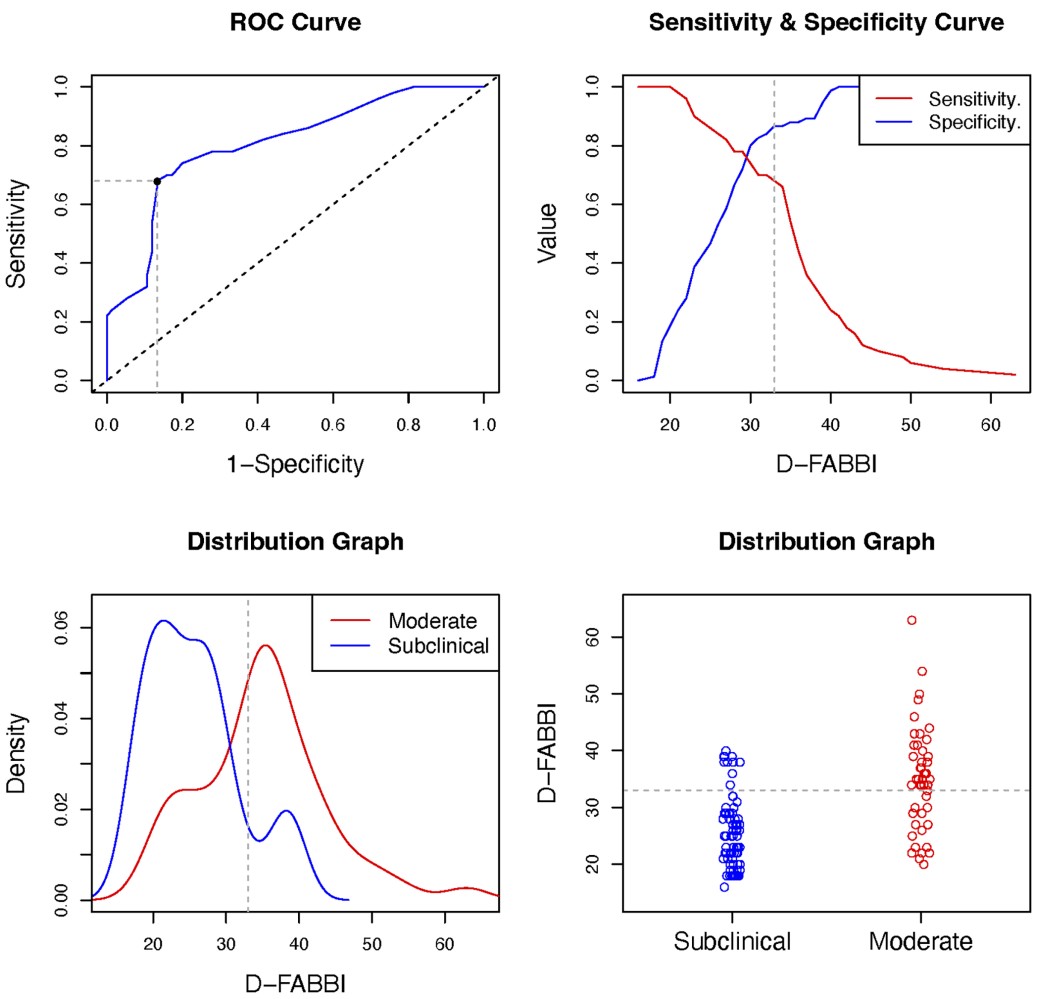

**Figure 4 Optimal cut-off point between levels of D-FABBI (Subclinical *vs.* Moderate).** A ROC (receiver operating characteristic) curve that represents the sensitivity of a diagnostic test that produces continuous results, depending on false positives (complementary to specificity), for different cut-off points, the image where the cut-off point at which the highest sensitivity and specificity is achieved and finally, a subclinical and moderate sample distribution graph.

The convergent validity results indicated that the D-FABBI with its two subscales presents moderate-strong positive associations with respect to the DHI and its subscales and low-moderate positive associations with respect to the HADS depression and anxiety subscales, as previously reported (*Dunlap et al., 2021b*; *Herdman et al., 2020b*). Although the highest association with DHI was 0.63, this might not be too high to be considered redundant, which is further evidence that the D-FABBI measures a variable other than disability, although it clearly has a relationship with this construct.

The instrument's behavior over a short period (7–8 days) indicates that it is a reliable (ICC > 0.80) and stable measure, with very low SEM and MDC values. Although the time elapsed between the test and retest can be considered very short, it is in agreement with the period recommended by *Streiner & Norman (2008)*, who suggested an optimal interval of 2–14 days between measurements. With such an interval, the tested participants would not

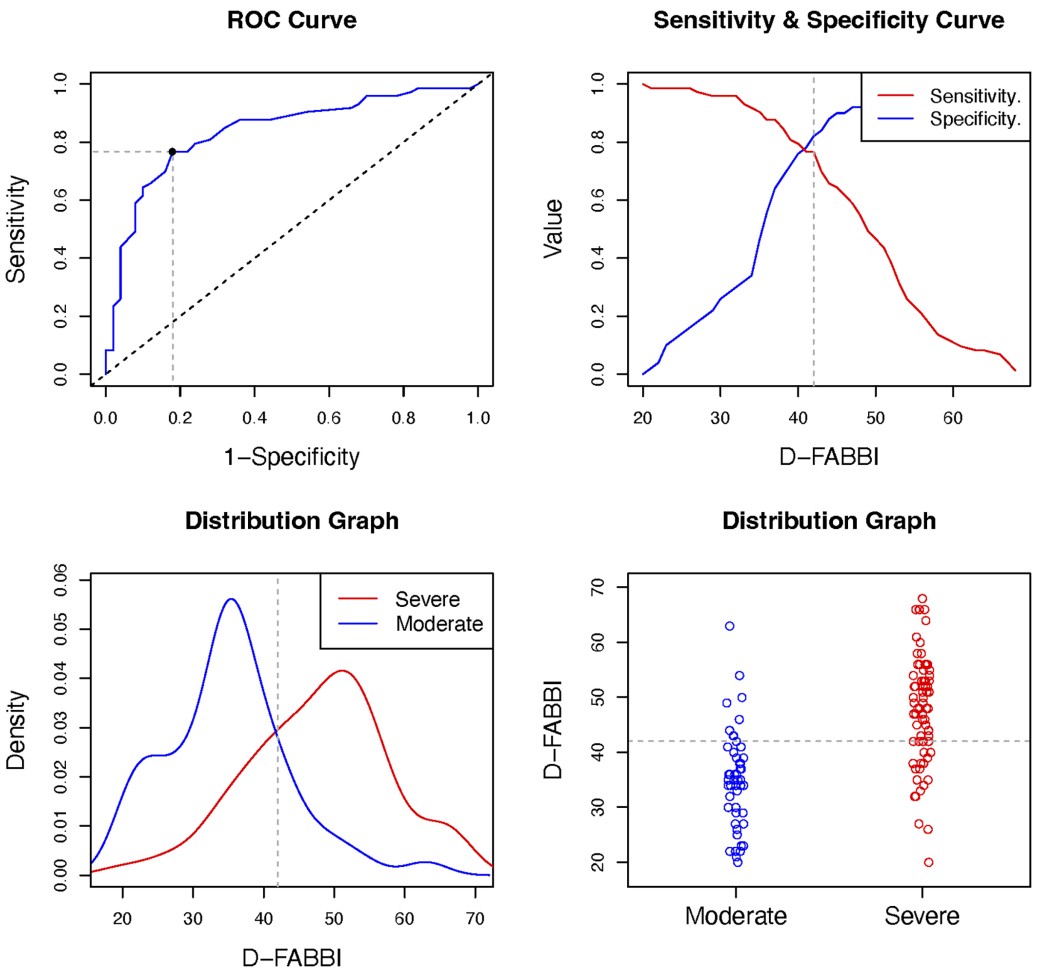

**Figure 5 Optimal cut-off point between levels of D-FABBI (Moderate *vs*. Severe).** A ROC (receiver operating characteristic) curve that represents the sensitivity of a diagnostic test that produces continuous results, depending on false positives (complementary to specificity), for different cut-off points, the image where the cut-off point at which the highest sensitivity and specificity is achieved and finally, a moderate and severe sample distribution graph.

remember their original answers, while not allowing sufficient time for the construct being tested to have changed (*Streiner & Kottner, 2014*).

The D-FABBI showed good sensitivity and specificity for discriminating patients with severe fear-avoidance behaviors and cognitions related to dizziness disability.
For moderate cognitions and behaviors, the specificity was good while the sensitivity was fair.

The results also indicated that patients with chronic vestibular disorders had higher scores on D-FABBI and on the two subscales than patients with acute and episodic vestibular disorders. It should be noted that the patients with chronic vestibular disorders presented greater severity of fear-avoidance behaviors and cognitions than the other two groups. A recent study found that patients with chronic vestibular disorders had higher levels of disability due to dizziness and higher rates of depression and anxiety as associated comorbidities than patients with episodic vestibular disorders (*Formeister et al., 2022*).

### Limitations

There are several limitations to the present study. First, the convergent validity of the instrument was compared with only two instruments that measure disability and anxiety and depressive symptoms. In future studies, it would be interesting to make comparisons with other self-report instruments that measure kinesiophobia and fear of falling.

Second, the discriminant validity analysis process compared only the results of D-FABBI between patients with various vestibular disorders. However, it might be necessary to analyze the behavior of the instrument with an asymptomatic population.

Lastly, this study analyzed only reliability, SEM and MDC. The instrument's behavior over time and when patients undergo vestibular rehabilitation treatment is not known. Future studies should identify the instrument's sensitivity to change and the clinically relevant change, considering interventions related to vestibular rehabilitation such as exercise, education, and cognitive-behavioral therapy.

## CONCLUSIONS

The results of this research indicate that D-FABBI is a bifactorial instrument with adequate validity, reliability, sensitivity and specificity to discriminate patients with vestibular disorders who present severe fear-avoidance behaviors and cognitions that might lead to disability.

The final version of the D-FABBI presented 17 items divided into a subscale measuring fear-avoidance behaviors and cognitions related to movement and another subscale related to ADL. Patients with chronic vestibular disorders had higher D-FABBI scores and greater severity compared with patients with episodic and acute vestibular disorders.

### Funding

The authors received no funding for this work.

### Competing Interests

Roy La Touche is an Academic Editor for PeerJ.

### Author Contributions

- Roy La Touche conceived and designed the experiments, analyzed the data, prepared figures and/or tables, authored or reviewed drafts of the article, and approved the final draft.
- Rodrigo Castillejos-Carrasco-Muñoz conceived and designed the experiments, performed the experiments, authored or reviewed drafts of the article, and approved the final draft.
- María Cruz Tapia-Toca conceived and designed the experiments, performed the experiments, authored or reviewed drafts of the article, and approved the final draft.
- Joaquín Pardo-Montero analyzed the data, authored or reviewed drafts of the article, and approved the final draft.

- Sergio Lerma-Lara analyzed the data, authored or reviewed drafts of the article, and approved the final draft.
- Irene de la Rosa-Díaz performed the experiments, prepared figures and/or tables, authored or reviewed drafts of the article, and approved the final draft.
- Miguel Ángel Sorrel-Luján performed the experiments, authored or reviewed drafts of the article, and approved the final draft.
- Alba Paris-Alemany conceived and designed the experiments, analyzed the data, authored or reviewed drafts of the article, and approved the final draft.

## Human Ethics

The following information was supplied relating to ethical approvals (*i.e.*, approving body and any reference numbers):

The study was approved by the bioethics committee of the Centro Superior de Estudios Universitario La Salle (CSEULS-PI-005/2020).

## Data Availability

The raw measurements are available in the Supplemental File 1. The raw shows all the data extracted from the participants regarding anthropometric, sociodemographic and inventory validation data.

## Supplemental Information

Supplemental information for this article can be found online at http://dx.doi.org/10.7717/peerj.15940#supplemental-information.

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
