# Peer review of "Development and validation of the dizziness fear-avoidance behaviours and beliefs inventory for patients with vestibular disorders"

_PeerJ, doi:10.7717/peerj.15940_

## Round 0.1 · original submission · Minor Revisions

Please revise according to the comments of these three reviewers. We hope they will re-review your revised manuscript after you have re-submitted. Thank you for your team's kind attention.

Reviewer 1 ·

Basic reporting

- The authors provided a clear and comprehensive description of the study design and outlined the recruitment process and inclusion criteria for participants, which were well thought out and justified based on the aims of the study.

- The design process of the proposed D-FABBI instrument and the experiment procedure was also well-described and included detailed information about the methods used.

Experimental design

- In the confirmatory factor analysis, the authors estimated both a CFA model with simple structure and an exploratory structural equation model (ESEM).The authors do not provide clear descriptions on the ESEM model. Adding more details on the models will provide enough information to readers.

Validity of the findings

- In the Exploratory Factor Analysis result section, 2 factors were finally determined. More details can be added to explain how the number of factors was decided. Either visualization results or analytical results will be informative.

- In the Confirmatory Factor Analysis result section, the authors mentioned modification indices were used to help identify areas where a model could be improved by adding or deleting a particular item from the measure. Since 4 items were reported, how did the authors determine the two MI models? What is the result if all 4 items are used instead of two items for each MI model?

Additional comments

- Line 310, replace “theorical” with “theoretical”.

Reviewer 2 ·

Basic reporting

Thank you for the opportunity to review this manuscript on the Development and validation of the Dizziness Fear Avoidance Behaviors and Beliefs Inventory for patients with vestibular disorders.

The research topic is both interesting and meaningful, and the author has done an excellent job in designing the study, performing analysis, and organizing the manuscript. The introduction is informative and includes important literature references in the field.

The table and figures are also well prepared

Overall, I find the manuscript to be acceptable, with only minor revisions required.

Experimental design

The method design was clear and reasonable, and the six-step procedure developed by the author was logical, comprehensive, and well-presented. The patient recruitment and exclusion were reasonable.

Suggestions:

In the Literature review subsection within the method section, references for each database could be very useful.

The choice of interpretative phenomenological analysis has more advantages like participant-centered and Transparency, which could also be mentioned in the context.

In the subsection on the development of items, the author mentions that 28 items were designed and subjected. It would be helpful to list these items to give the reader a better understanding of the content and scope of the instrument if there’s enough space.

Validity of the findings

The authors did a comprehensive job of analyzing the data and evaluating the validity, accuracy, and reliability of the proposed D-FABBI.

Suggestions:
At the beginning of the Result section, a table for the one-factor ANOVA with Bonferroni correction could be helpful for the reader to understand the results.

In terms of table formatting, it may be advisable for the author to avoid using bold text for certain words.

With regard to the figures, I recommend that the author maintain consistent font styles across all figures and ensure that any textual information contained within the figures is legible and easily comprehensible.

Additional comments

No comment

Reviewer 3 ·

Basic reporting

In this manuscript, the authors design a new methodology to measure fear-avoidance behaviors's association with vestibular disorders. The method used here is two-fold - a longitudinal study with N=35 initial participants, as well as a literature review of papers in this field.

The English is clear and tables/figures are amply provided

Experimental design

I would advise the authors to provide more details on the specific numbers they choose for thresholding or design of their methodology, e.g.,

* Line 242: How were the exclusion criteria measured?
* Line 258-259: How were the minimum acceptable and expected values of ICC determined?
* Line 306: How was the threshold for factor loading, 0.4, determined


Minor comment:
* In Line 176, it is customary to provide the initials of the two authors who evaluated the content analysis

Validity of the findings

The findings seem valid. I appreciate that the authors correctly accounted for the critical values using Bonferoni corrections

---

## Round 0.2 · accepted · Accept

The manuscript has been accepted.

Reviewer 1 ·

Basic reporting

I think that the authors have adequately addressed the comments made by the reviewers in the revised version of the manuscript. Therefore, I have no further comments.

Experimental design

I think that the authors have adequately addressed the comments made by the reviewers in the revised version of the manuscript. Therefore, I have no further comments.

Validity of the findings

I think that the authors have adequately addressed the comments made by the reviewers in the revised version of the manuscript. Therefore, I have no further comments.

Reviewer 2 ·

Basic reporting

Thank you for inviting me again to review the revised manuscript on the Development and validation of the Dizziness Fear Avoidance Behaviors and Beliefs Inventory for patients with vestibular disorders.

The authors did a good job of addressing the reviewers’ comments. I did find the manuscript to be acceptable for publishing.

Experimental design

Key references for each database were added. The author chose not to present all 28 items that are reasonable. I think the design of the experiment is well introduced.

Validity of the findings

The author mentions that the information in Figure 2 is enough instead of an ANOVA table. I agree with them. It’s very informative.

All figures are well-adjusted.

Reviewer 3 ·

Basic reporting

The authors have addressed my comments and I can now accept this manuscript for publication

Experimental design

NA

Validity of the findings

NA